# Magnetic Domain Transition of Adjacent Narrow Thin Film Strips with Inclined Uniaxial Magnetic Anisotropy

**DOI:** 10.3390/mi11030279

**Published:** 2020-03-08

**Authors:** Tomoo Nakai

**Affiliations:** Industrial Technology Institute, Miyagi Prefectural Government, 2-2 Akedori, Aoba-ku, Sendai-city, Miyagi 981-3206, Japan; nakai-to693@pref.miyagi.lg.jp; Tel.: +81-22-377-8700

**Keywords:** stepped magneto-impedance element, magnetic domain, domain transition, distributed field, normal field, many-body elements, thin film

## Abstract

This study deals a phenomenon of magnetic domain transition for the stepped magneto-impedance element. Our previous research shows that an element with 70° inclined easy axis has a typical characteristic of the domain transition, and the transition can be controlled by the normal magnetic field. In this paper, we apply this phenomenon and controlling method to the line arrangement adjacent to many body elements, in which mutual magnetic interaction exists. The result shows that the hidden inclined Landau–Lifshitz domain appears by applying a distributed normal field the same as an individual element.

## 1. Introduction

The observation of a magnetic domain of a thin film magnetic element is carried out using measuring methods such as electron beam [1], Kerr magneto-optic effect [2], scanning tunneling microscope, and magnetic force microscopy. Recent topics of investigations on the domain structure range from the nanoscale to the submillimeter-scale, such as a formation of vortices [3], an effect caused by ion irradiated sheet [4] and a property improvement of the silicon steel magnetic core [5]. The structure of the magnetic domain formed in a thin film element having a dimension from several to hundreds of micrometers was previously reported, showing that it forms several typical domain patterns, the Landau-Lifshitz domain [6], for example, and the variations as a function of the strength of the external field [7]. For the research field of soft magnetic thin films, the domain structure has been studied in accordance with the study of magneto-impedance magnetic field sensors. The recent performance improvement of an apparatus of the magneto-optic Kerr effect (MOKE) has contributed to the investigation of the magneto-impedance (MI) sensor from the viewpoint of the dependence of high-frequency impedance on the magnetic domain structure of the element. The beginning of it seemed to be a physical investigation of the MI sensor for clarifying the sensing mechanisms [8,9,10]. Thereafter, the study of the magnetic domain for thin film MI sensors has been developing continuously for the purpose of improving the properties of the MI sensor based on the control of the magnetic domain. The following are the typical example of the study: A layered and laminated structure of thin film element [11,12,13], a miniaturization of the element [14], a consideration of magnetostriction [15], an effect of high-temperature annealing [16], an effect of element dimension for a magnetic property such as coercivity *H*c [17], direct current (DC) biasing [18], and a biasing caused by the exchange force [19]. A domain structure simulation was originally developed for the methods which were suitable for the actual size of the MI element which is unable to apply the conventional micro-magnetic simulations due to larger dimensions of the element [20,21].

The stepped-MI element has the unique property of a step-like magneto-impedance change, in the case where the sensor has an in-plane uniaxial inclined easy axis [22,23]. A domain observation shows that the step-like change is due to a magnetization transition within three states, including the longitudinal single domain with parallel state, with an anti-parallel state and the inclined Landau–Lifshitz domain (ILLD). This phenomenon is expected to create a sensor with a memory function [24]. In a condition where the sensor has an easy axis of 70° relative to the short-side axis of the rectangular element, the transition is limited between the parallel and anti-parallel states despite the existence of a stable ILLD [25]. We call this stable ILLD state as a hidden ILLD state. The hidden ILLD state is able to be appear by applying a normal magnetic field with a distributed inclined angle. The normal direction is defined as a surface normal direction to the substrate plane, and the inclination angle of the field is defined as an angle between the normal direction and the direction of the magnetic field having a certain inclination toward the length direction of the element. The distribution of the inclination means a variation of the inclination angle as a function of the length position of the element. The appearance of the hidden ILLD state was estimated by numerical analysis [21,26,27] and experimentally confirmed [28,29] using a single element. The artificial transition from the single domain state toward the hidden ILLD state was also realized experimentally [30]. It is expected to apply to an unerased memory and its reset procedure.

In this study, an extension of the application of this phenomenon to clustered many-body elements was experimentally investigated. The individual element has the same width and thickness, and also has the direction of magnetic anisotropy the same as an element having the stepped-MI property with a hidden ILLD state. A certain number of the elements were arranged in a planar cluster which has a configuration of a line arrangement adjacent to many-body elements. This trial is important for realizing a high-density device using this phenomenon. The magnetic hysteresis loop (MH-loop) of the planar clustered element was measured in accordance with a domain observation. An effect of the application of the distributed normal field was also investigated.

## 2. Experimental Procedure

The element was fabricated by a thin-film process. An amorphous Co_85_Nb_12_Zr_3_ film was RF-sputter deposited onto a soda glass substrate and then micro-fabricated into rectangular elements by a lift-off process. The element was thousands of µm in length, 20 µm wide, and 2.1 µm thick. A uniaxial magnetic anisotropy was induced by magnetic field annealing, 240 kA/m at 673 K for 1 h. The easy axis of the magnetic anisotropy is made along the processing magnetic field. The annealing apparatus used in this study had an accuracy in the angle position of 0.5°. In this study it was induced in several different conditions for a comparison of their properties.

We made two different layouts of the elements. One was the configuration of the line arrangement adjacent many-body elements having a mutual magnetic interaction with each other, and the other layout were dispersed individual elements. These two types of different layouts were made for a comparison between the elements having the mutual force and those without it. The prior many-body element was prepared in three different conditions for the purpose of confirming the effect of the direction of easy axis with a focus on the element having the hidden domain state, which is around *θ* = 70°. The element dimensions were as follows: In the case of the adjacent many-body elements, the length was 3000 µm, the width was 20 µm, and the thickness was 2.1 µm. The element length was determined by a knowledge obtained by the previous study that the element would have a residual domain at the both end of element. In our previous report, the element length was set as 2000 µm [29]. This element was assembled to form a parallel line arrangement configuration with the line and space (L/S) as 20 µm and 20 µm. The two dimensional area of this assembled elements was 3000 µm × 3020 µm, which was suitable for a measurement of magnetization loop using a vibrating sample magnetometer (VSM). The three different directions of easy axis were made in *θ* = 61°, *θ* = 71°, and *θ* = 90°. Our previous study showed that the element of *θ* = 61° has a changing property from single domain, −, to the ILLD and then to single domain, +, with the increasing magnetic field. The element of *θ* = 90° has a changing property between the single domain − and +. These are the neighboring conditions in the direction of easy axis in *θ* = 71°, which is the focusing condition in this study. The element layout on the glass substrate is shown in Figure 1. A schematic explanation of the direction of the easy axis and also an enlarged view of a part of a fabricated element is shown in Figure 2. On the other hand, the latter layout, which has dispersed individual elements, consisted of 54 elements on a 26 mm × 26 mm glass substrate. The distance of each element was set as follows: A longitudinal distance of 2000 µm and a lateral distance of 2500 µm. The dimensions of each element was as follows: A length of 2000 µm, a width 20 µm, and a thickness 2.1 µm. The angle of the magnetic easy axis was directed in *θ* = 67° against the width direction of the element strips. This layout on the glass substrate is shown in Figure 3. The element with *θ* = 67° was expected to have a property of the three stable states [24] with a narrow multi-domain range.

The measurements carried out here were a MH-loop measurement and a magnetic domain observation. The MH-loop was measured by a vibrating sample magnetometer (VSM) (TM-VSM211483-HGC, TAMAGAWA Co., Ltd., Sendai, Japan). The magnetic domain was observed by a Kerr microscope (BH-762PI-MAE, NEOARK Corporation, Tokyo, Japan) by applying a magnetic field in a certain direction and also at a certain controlled strength. These applied magnetic fields were generated by the following two apparatuses. The distributed normal field was made using a ring-shaped magnet which was positioned below the observation stage of the Kerr microscope. The magnetic field along the element’s longitudinal in-plane direction was controlled using a Helmholtz coil (Custom-made one, Ryowa Electronics Co., Ltd., Sendai, Japan) while the observation of the magnetic domain was carried out. Figure 4 and Figure 5 show the schematic and photo of the measurement apparatus of the magnetic domain used in this study.

The definition of the distributed normal field is shown in Figure 6, which is the same definition as the previous article [29].

## 3. Results

A photo of the fabricated many-body elements on a glass substrate are shown in Figure 7. Each element was divided individually by a dicing apparatus and then used for the measurements. The front three elements of the photo were the measured samples of the L/S = 20/20 µm layout. The measurements in this study were carried out using this element. The results are shown in the following subsections.

### 3.1. Magnetization Process of the Many-Body Elements

Firstly, the MH-loop measurement for the many-body elements with different directions of the easy axis was carried out. The length direction of the element line was set in the measurement direction of the VSM apparatus. Measured MH-loops are shown in Figure 8. Figure 8a is a case for the easy axis in *θ* = 61°, Figure 8b for *θ* = 71°, and Figure 8c for *θ* = 90°. The measurement sweep speed was 0.1 Oe/s (7.96 (A/m)/s) and the measurement time constant was 100 ms. The MH-loop in each condition resembles each other. The coercivity slightly increased with the increase in the *θ* value, and the linearly-inclined part of the curve changes to a slightly bending configuration in the vicinity of the zero field with the increase in the *θ* value. In these figures, the value of *M*_eff_ (T) is estimated from the volume 3000 μm × 3020 μm × 2.1 μm. This is an effective value, because it is estimated as if the element has a shape of a single square sheet which corresponds to the volume of many-body elements as a whole.

The observation of the magnetic domain was also carried out and shown in Figure 9. The MH-loop and corresponding photo of the magnetic domain at a certain magnetic field is shown in the figure. Note that the magnetic domain observation at different strengths of the magnetic field was carried out individually, not sequentially. The MH-loop measurement was a sequential measurement, on the other hand. Figure 9a is the case for the easy axis at *θ* = 61°, Figure 9b for *θ* = 71°, and Figure 9c for *θ* = 90°. The elements were magnetically saturated in the value around 300–400 A/m. Based on our previous study the variation of the magnetic domain for the individual single element, having an easy axis at *θ* = 61°, has a property of domain variation which changes from a single domain in the − direction to ILLD around zero magnetic field, and then a single domain in the + direction, with the increase in the magnetic field from minus value to the plus value. In the case of *θ* = 71°, it is the element having the hidden-stable state. This means that the element has an energetically stable ILLD state, however, it does not appear using a method of application of a uniform external field. Apparently the domain switched between the − single domain and the + single domain for the *θ* = 71° element. In the case of *θ* = 90°, the domain simply switched between the − single domain and the + single domain. The measured result of this study, as shown in the Figure 9, has a changing property of each element but not all at once. In our fabrication process, the magnetic property of the thin film on a glass substrate was a uniform one, therefore, the domain transition should occur simultaneously. However, the result shows that it does not occur all at once. It is expected that it comes from the magnetically mutual interaction between the accumulated many-body elements. Each element in a many-body element has a certain magnetization, therefore, each element has a certain value of the magnetic pole at the end of the narrow strip. The magnetic pole of an element makes an inverse magnetic field to the surrounding elements. According to the applied external field the number of transited elements changes, as shown in Figure 9. A magnetic field which was felt in a certain element was a sum of the field caused by other elements in a many-body and the external field. Therefore, it has a certain inverse effect of the magnetic field as if there exists a well-known “demagnetizing field”. Based on these three experimental results the magnetic domain transition of different easy axis conditions is explained as follows:(a)*θ* = 61°: The initial single domain in the − direction of whole elements as shown in the photo view change to the ILLD gradually and individually with the increase in the applied field. After the domain of all elements changes to the ILLD, then the width of the area of the striped domain gradually changes and, finally, the stripe of the ILLD is erased and all elements change to the single domain in the + direction. When the elements are all in the state of a single domain, +, the many-body element is getting magnetically saturated.(b)*θ* = 71°: The initial single domain in the − direction of whole elements as shown in the photo view change to the single domain in + gradually and individually with the increase in the applied field. When the elements are all in the state of single domain +, the many-body element is getting magnetically saturated.(c)*θ* = 90°: The domain transition was the same as the one with *θ* = 71°.

From the view point of the magnetic domain, both Figure 9b and Figure 9c had, qualitatively, the same profile of domain transition. This is explained based on the characteristic of the domain transition of individual elements. The individual domain transition was the switched property between + and − single domains. The hidden ILLD in Figure 9b did not appear even in the case of many-body configuration. It is the same as the case of the individual element. The domain transition in Figure 9a is clearly different from the other two conditions. In this case it is characterized by the appearance of the ILLD around zero external field.

Figure 10 shows a variation of element number of the appearance of the domain transition as a function of the applied magnetic field. It is plotted based on the number of transited elements in the middle area of the many-body rectangular plane. Figure 10a shows a variation for the case of easy axis in *θ* = 71°, and Figure 10b shows the case for *θ* = 90°. The measured MH-loop in Figure 8 must be a whole sum of the magnetization of individual elements in the many-body element. Then the plot of Figure 10 would expect to match with the MH-loop in Figure 8. Actually, both profiles are in good agreement, especially the appearance of bending property of the magnetization curve in the vicinity of the zero field.

### 3.2. Comparison with Individual Element

For the purpose of discovering the effect of magnetic mutual interaction, a comparison of the magnetization process between the adjacent many-body element and the dispersed individual elements was carried out based on the variation of the domain transition number as a function of the applied field, as plotted in Figure 11. This transition was counted based on an observed domain transition of the individual element of the fabricated elements of the layout as shown in Figure 3.

Figure 12 shows a detailed domain transition for an individual element as a function of applied field. The applied field was in the in-plane length direction. This *θ* = 67° element has a splitting appearance of the ILLD as a function of the applied field. When the field is increasing the − single domain switched to the ILLD in the + region of the applied field, and it switches within the − region when the field is decreasing. This profile was already reported and explained previously [24]. In Figure 11 the number of transitions was counted when the domain changed from the single domain to the ILLD in both directions of the changing field.

Based on Figure 11 and Figure 12, the domain transition of the individual elements, which was fabricated on the same substrate, occur at almost the same value of the magnetic field. It is derived from the experimental phenomenon that the transition occurs sharply in the vicinity of the zero field, as shown in Figure 11. With the comparison of this result and Figure 10a, the gradual transition of the domain is explained as a result of the magnetic mutual interaction.

### 3.3. Effect of Distributed Normal Field

From here an examination of the reconstruction of the hidden ILLD is shown. The method used here was previously developed by us [29] for the case of a single element. It is the method of application of a distributed normal field. In this paper a trial for the many-body element is investigated for the first time. Prior to this paper the many-body element having the easy axis at *θ* = 71° was experimentally shown to behave as if it is the individual element having the same easy axis direction. The domain of the consisted each element switched as expected from the behavior of the individual single element with hidden ILLD state, whereas the domain switching does not occur simultaneously within the many-body element. This dispersed behavior in the switched field would be expected to come from the mutual magnetic interaction. In this section, an application of the distributed normal field was tried for the *θ* = 71° many-body element.

Figure 13 shows a vector distribution of magnetic flux density generated by a ring-shape magnet. This magnet was used for generating the distributed normal field in the apparatus of domain observation, Kerr microscope. The dimensions of the NdFeB ring-shape magnet were as follows: The outer diameter was 30 mm, the inner diameter was 15 mm, the thickness was 2 mm. The magnetic poles were placed both on the upper and a bottom surfaces of the ring-disc. The vector distribution shows that there are two phases, one is an area in which the magnetic flux forms a closed circular configuration with a small diameter and the other is an area in which the vectors bound for or from outside having almost parallel configuration. The prior was placed in the near of the magnet and the latter was relatively far from the magnet. In this study, the many-body element was placed in the area of almost parallel vector which was relatively far from the magnet.

Figure 14a shows a variation of *B_x_* as a function of the X-position at different distances from the magnet, *d* = 8.5 mm and *d* = 10 mm. “*d*” was defined in Figure 4. The position *x* = 0 was placed on the center axis of the ring-shaped magnet, and the measured element was placed on the X–Y plane at a certain Z-position, which means different d-positions in this measurement. The variations in the figure show linear configurations having different inclination, one is *B_x_* (mT) = 0.54 × *x* (mm) at *d* = 10 mm and the other is *B_x_* (mT) = 1.22 × *x* (mm) at *d* = 8.5 mm. The normal field *B_z_* as a function of X-position is shown in Figure 13b. The normal field at *x* = 0 was 11.5 mT at *d* = 10 mm and 8.7 mT at *d* = 8.5 mm. The normal field in the element’s rectangular area varies at most 8%, therefore, it has a slight difference from the ideal variation (see Figure 6).

The result of the application of the distributed normal field to the element having the easy axis in *θ* = 71° is shown as follows: Figure 15 shows a magnetic domain at *B_x_* = 0, without applying a distributed field. Figure 16a shows a magnetic domain when the *d* = 10 mm in which the distributed parameter *∆B_x_/∆x* = 0.54 (T/m), and Figure 16b shows a magnetic domain when *d* = 8.5 mm in which the distributed parameter *∆B_x_/∆x* = 1.22 (T/m). These results show that the ILLD state appears in the central area of the many-body element when the distributed normal field is applied. These ILLD have a certain periodical pattern in the element lines in spite of the randomly-switched single domain appearance at the zero field without the distributed normal field. This periodical pattern has a variation as a function of the distributed parameter.

## 4. Discussion

A consideration concerning the magnetic mutual interaction within the many-body element is discussed here. A qualitative estimation of the interaction is carried out based on a magnetic field distribution originating from one narrow magnetic strip which is a construction element of the many-body element.

Figure 17 shows a vector diagram of magnetic flux density arising from one narrow strip element. We call the magnetic flux density the “magnetic field”, the same as previous section. The analysis is done using a 3D FEM simulation of the static magnetic field. In this analysis the single narrow element having a magnetic single domain in the longitudinal direction was modeled as one which was magnetically saturated along the X+ direction with a value of saturation magnetization as *M_s_* = 0.93 T. This *M_s_* value was the measured value of Co_85_Nb_12_Zr_3_ amorphous thin film. Figure 17 is the vector distribution on the substrate plane which comes from the saturated narrow strip element. This figure indicates a simple directional distribution diagram, therefore, the length of the arrow does not show the strength of the magnetic field. It is easily presumed that the distribution forms a magnetic field arising from one source and one sink configuration. In this case both the source and the sink are placed at the end of the narrow element. The simulation layout was set as the element axis was placed along the X axis and the center position of the element was set at the origin. A distance from the element is shown as *D_e_*.

Figure 18 shows the magnetic field distributions along the axis of elements in the neighboring position of the line arrangement many-body element. The L/S = 20/20 μm configuration made the distance of axes to be the multiples of 40 µm. In this figure, variations of magnetic field *B_x_*, which is the field in the direction of longitudinal axis is shown as a parameter of element distance *D_e_*. The horizontal axis of the figure shows a longitudinal position x on the axis of the neighboring virtual element. The direction of magnetic field *B_x_* has a negative value, due to the magnetized direction of the existing element being positive, as shown in Figure 17. For the reason of easy understanding the vertical value was made as an absolute value of *B_x_*. The element distance *D_e_* ranges from 40 µm to 200 µm, which corresponds to the position from the just neighboring to the 5th neighboring element. Figure 18 shows that the |*B_x_*| has a maximum value in the vicinity of the end of the element, the range was almost 0.3 mm from the end, and the maximum value rapidly decreased with the increase in the *D_e_* value. From Figure 12 the magnetic domain transition appears around 500 mOe (0.5 × 10^−4^ T). The magnetic field which exceeds this value would expect to have a promoting effect of domain transition. The inverse field generated by a transited element have a blocking effect for the neighboring elements. The effect is expected to be strongly affected to the 1st, 2^nd^, and 3rd neighboring elements.

The magnetic field distribution in Figure 17 also shows that a certain weak field widely spread around the single element. The field ranges more than 1.5 mm. Figure 19 shows the variation of magnetic field *B_x_* on the Y-axis of Figure 17 as a function of the distance from the element *D_e_*. The *B_x_* monotonically decreases as a function of *D_e_*, such as *B_x_* = 2.7 × 10^−6^ T (27 mG) at *D_e_* = 20 µm and *B_x_* = 1.0 × 10^−6^ T (10 mG) at *D_e_* = 1.5 mm. This magnetic field is much smaller than the maximum value of the one in Figure 18, whereas when we consider a cumulative effect of the many-body element which has 76 adjacent lines in an element, the sum of the magnetic fields would be expected to exceed the transition field 500 mOe (0.5 × 10^−4^ T).

According to the experimental observation in this study, the blocking effect of the domain transition in the neighboring element of a transited single narrow element was observed, as can be seen in Figure 9. The measured MH-loop of the many-body element, as shown in Figure 8, also suffered an effect as if it suffered a demagnetizing field. These effects would be considered to come from both the short-range effect of magnetic field generated at the edge of a single narrow element, and also the cumulative effect based on the accumulation of the widely spread weak field within the whole area of adjacent many-body elements.

## 5. Summary

An investigation of the stepped-MI phenomenon for the clustered many-body elements was experimentally carried out. The 76 elements were arranged in a planar cluster with the line/space = 20 µm/20 µm having a configuration of the line arrangement adjacent to many-body elements. The individual element in it has the same width and thickness, and also has the direction of magnetic anisotropy the same as an element having the stepped-MI property with a hidden ILLD state, which is *θ* = 71°. The MH-loop of the planar clustered element was measured in accordance with a domain observation. The MH-loop of the clustered many-body element which has the easy axis in *θ* = 71° was almost the same as the one having the easy axis in *θ* = 61° and *θ* = 90°. The each element has suffered a mutual magnetic interaction between the consisting element strips, as if it suffered the demagnetizing force. The variation of magnetic domain for the *θ* = 71° many-body element was a gradually changed from the number of switched elements having the single domain, and which is the same as the domain variation of *θ* = 90°. An effect of the application of the distributed normal field was also investigated. The ILLD appeared by applying the distributed normal field. A periodicity of the domain distribution formed by the consisting elements was observed, and the periodical pattern changed as a function of the distributed parameter *∆B*_x_/*∆x*.

## Figures and Tables

**Figure 1 micromachines-11-00279-f001:**
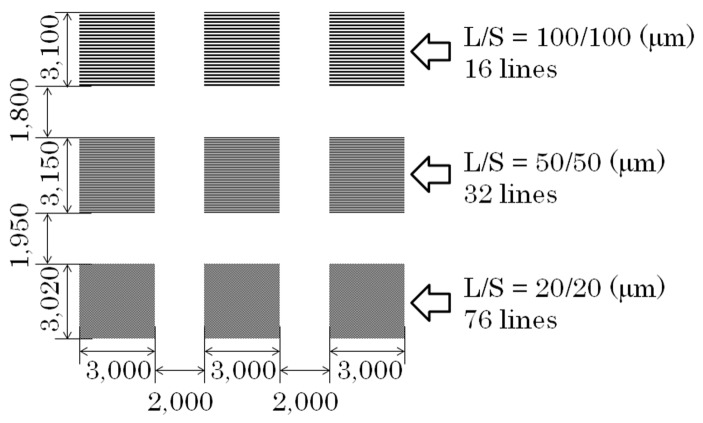
Layout of the many-body element on a glass substrate.

**Figure 2 micromachines-11-00279-f002:**
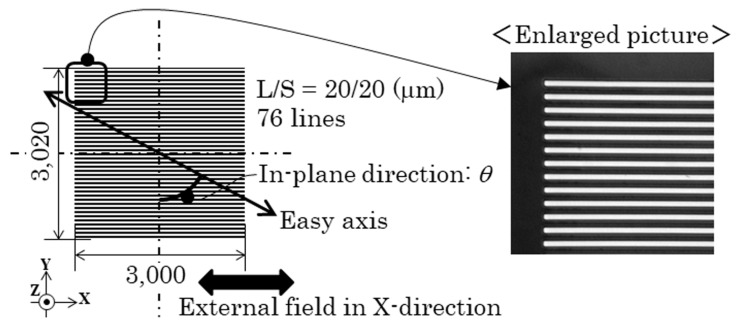
Direction of easy axis *θ* and an enlarged picture of the fabricated element.

**Figure 3 micromachines-11-00279-f003:**
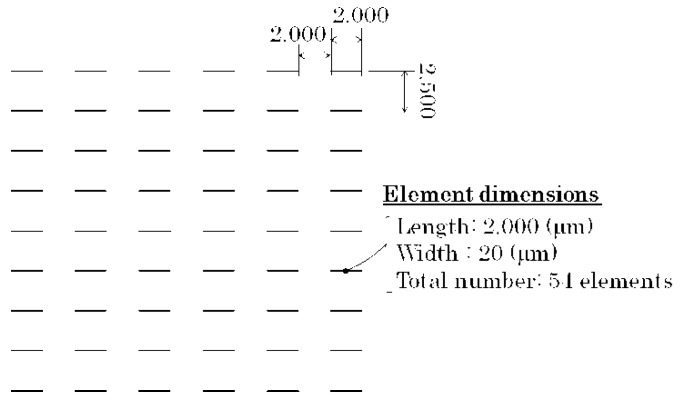
Layout of dispersed individual elements on a glass substrate.

**Figure 4 micromachines-11-00279-f004:**
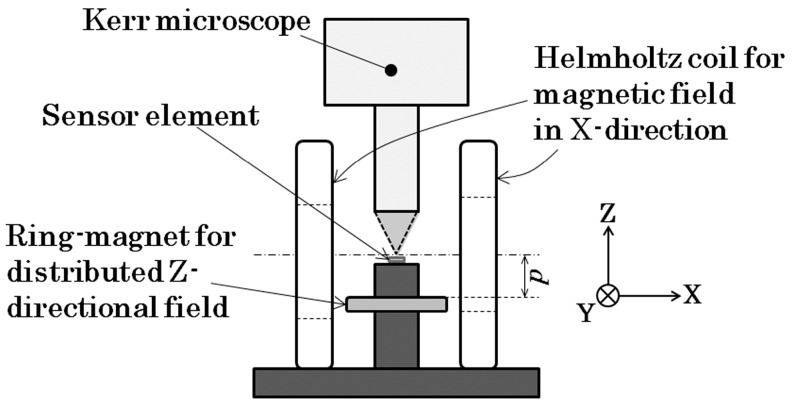
Schematic of the measurement apparatus.

**Figure 5 micromachines-11-00279-f005:**
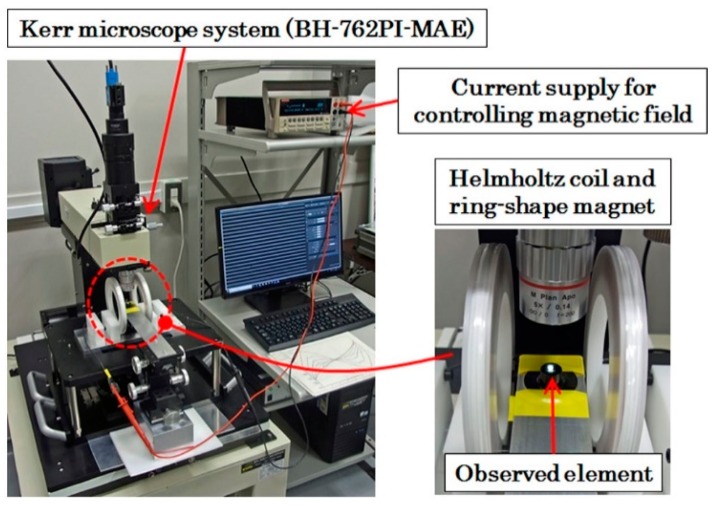
Photo of the measurement apparatus.

**Figure 6 micromachines-11-00279-f006:**
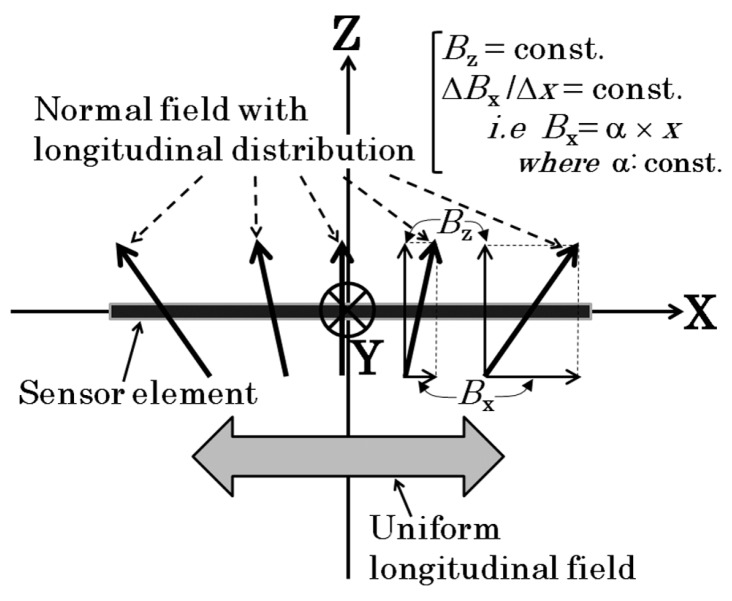
Definition of the distributed normal field.

**Figure 7 micromachines-11-00279-f007:**
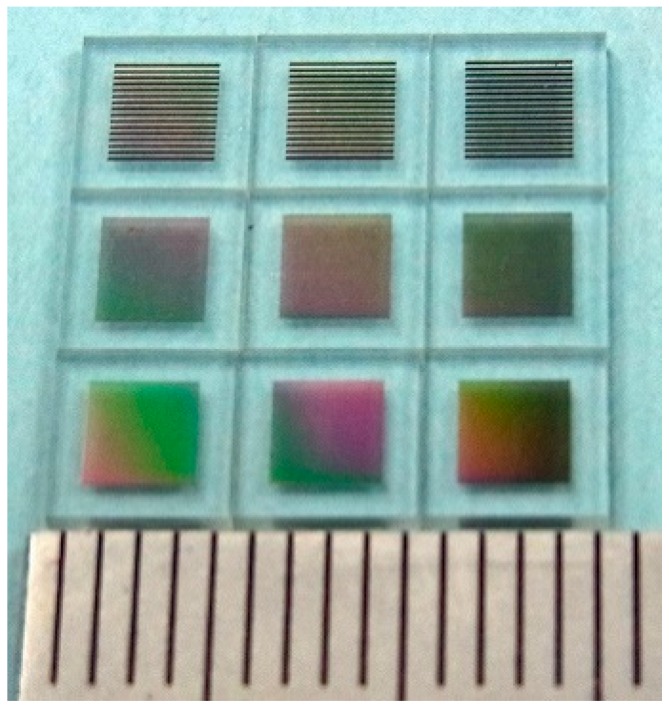
View of the fabricated many-body elements.

**Figure 8 micromachines-11-00279-f008:**
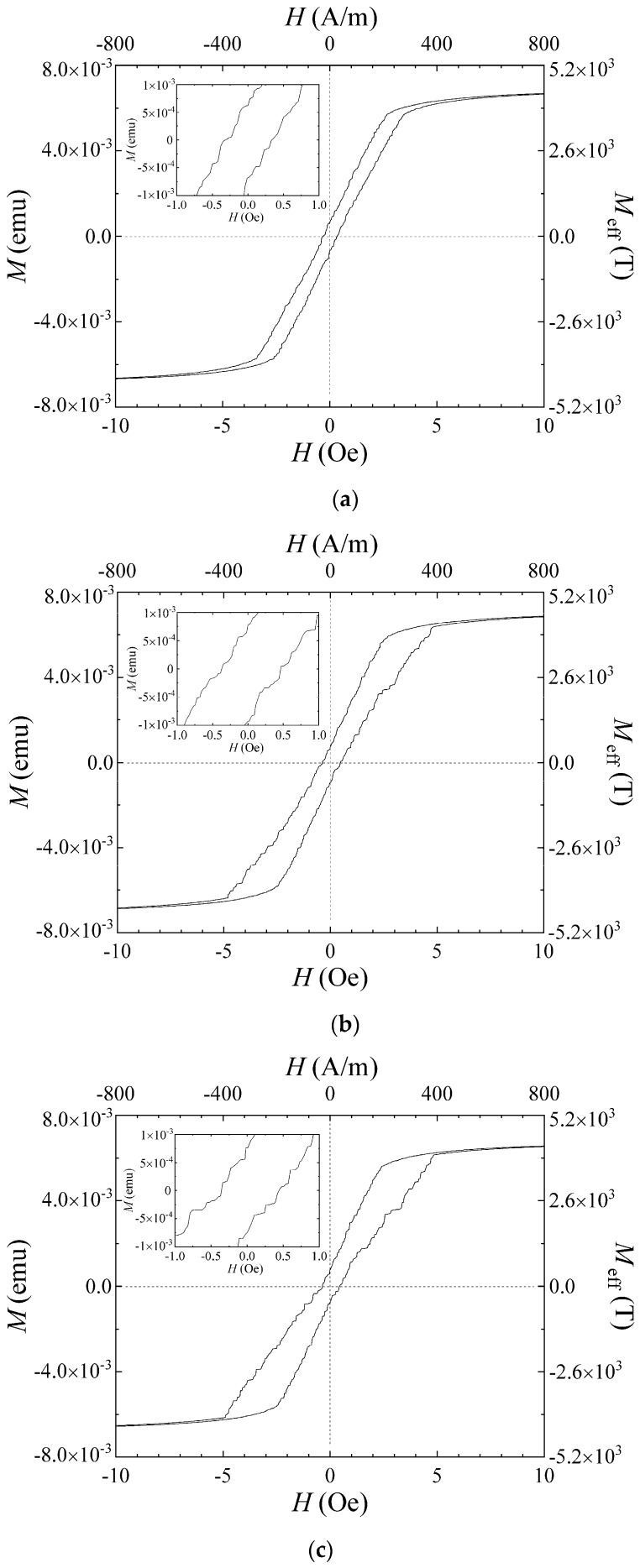
MH-loop of the fabricated elements. (**a**) The case for easy axis *θ* = 61°. (**b**) The case for easy axis *θ* = 71°. (**c**) The case for easy axis *θ* = 90°.

**Figure 9 micromachines-11-00279-f009:**
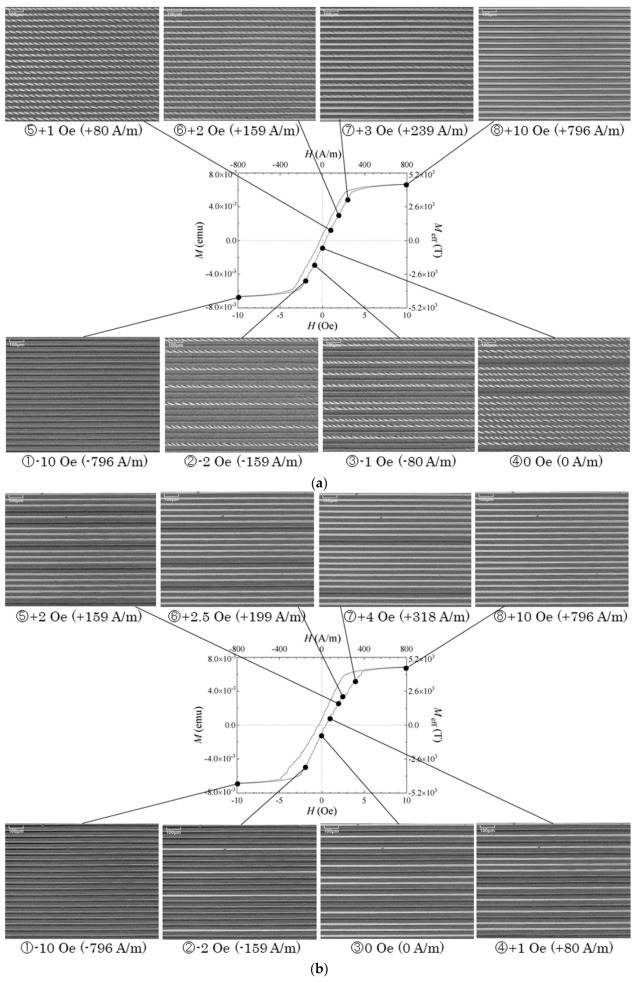
MH-loop and corresponding photo of the magnetic domain. (**a**) The case for easy axis *θ* = 61°. (**b**) The case for easy axis *θ* = 71°. (**c**) The case for easy axis *θ* = 90°.

**Figure 10 micromachines-11-00279-f010:**
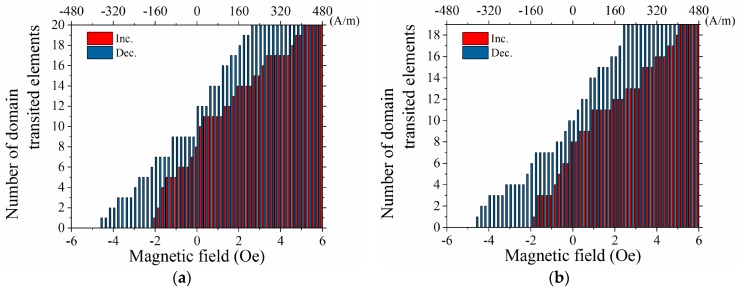
Number of domain transitions as a function of applied magnetic field. (**a**) The case for easy axis *θ* = 71°. (**b**) The case for easy axis *θ* = 90°.

**Figure 11 micromachines-11-00279-f011:**
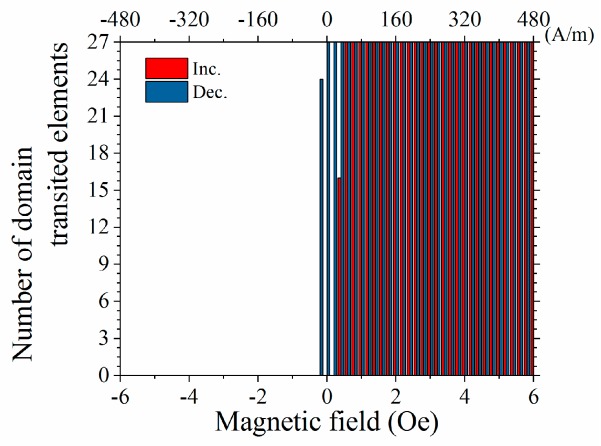
Number of domain transitions as a function of the applied magnetic field for individual elements.

**Figure 12 micromachines-11-00279-f012:**
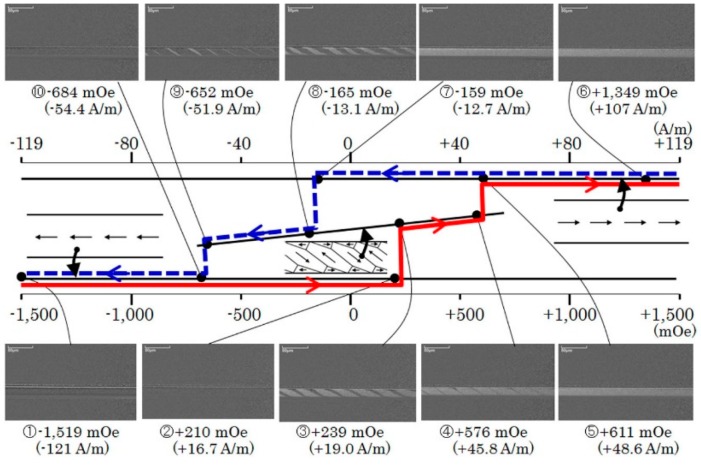
Detail of the domain transition for an element of easy axis *θ* = 67°.

**Figure 13 micromachines-11-00279-f013:**
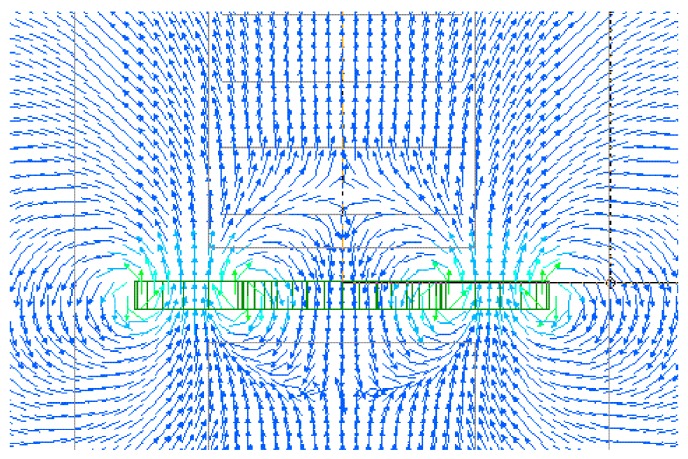
The vector of the magnetic flux density generated by a ring-shaped magnet.

**Figure 14 micromachines-11-00279-f014:**
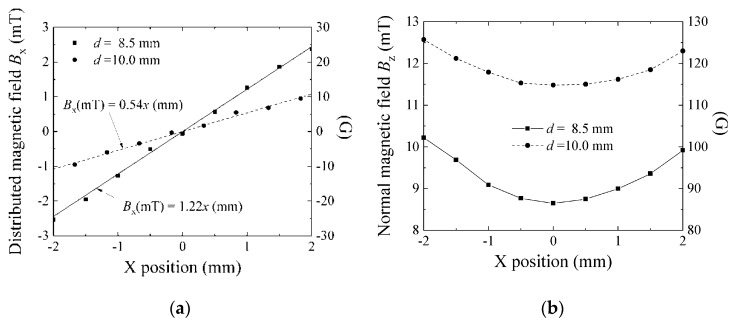
The distribution of the magnetic field generated by a ring-shaped magnet. (**a**) Variation of *B_x_* as a function of *x.* (**b**) Variation of *B_z_* as a function of *x*.

**Figure 15 micromachines-11-00279-f015:**
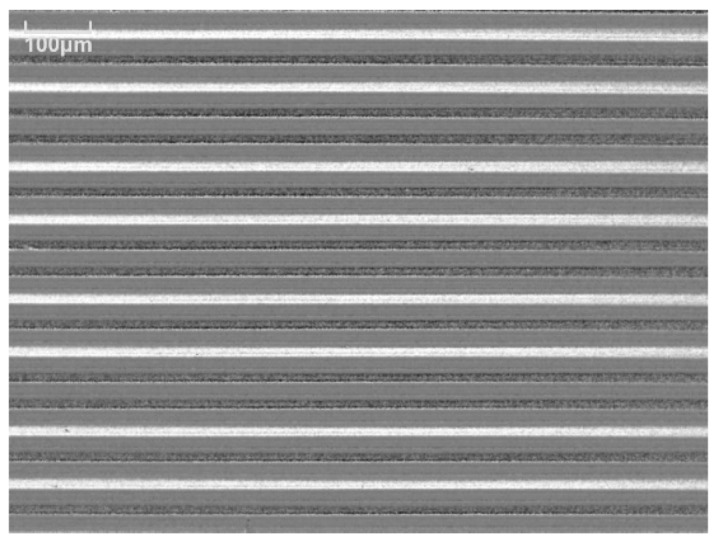
Magnetic domain without applying distributed field (*θ* = 71°).

**Figure 16 micromachines-11-00279-f016:**
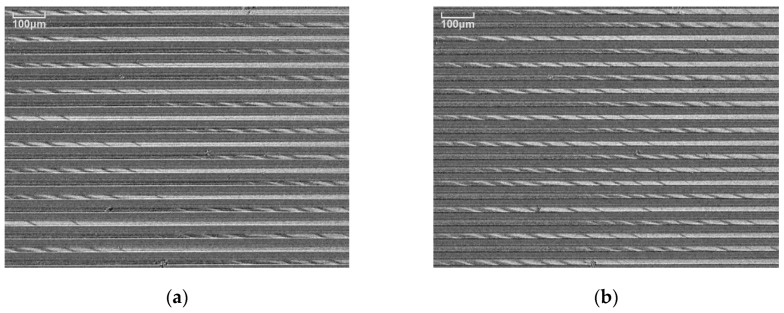
Magnetic domain with applying distributed field (*θ* = 71°). (**a**) Distributed parameter *∆B_x_/∆x* = 0.54 (T/m). (**b**) Distributed parameter *∆B_x_/∆x* = 1.22 (T/m).

**Figure 17 micromachines-11-00279-f017:**
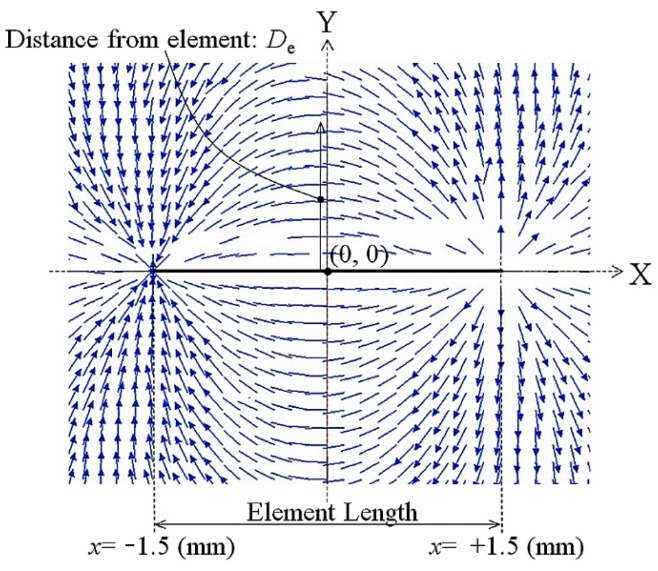
Vector diagram of the magnetic flux density arising from one narrow strip element.

**Figure 18 micromachines-11-00279-f018:**
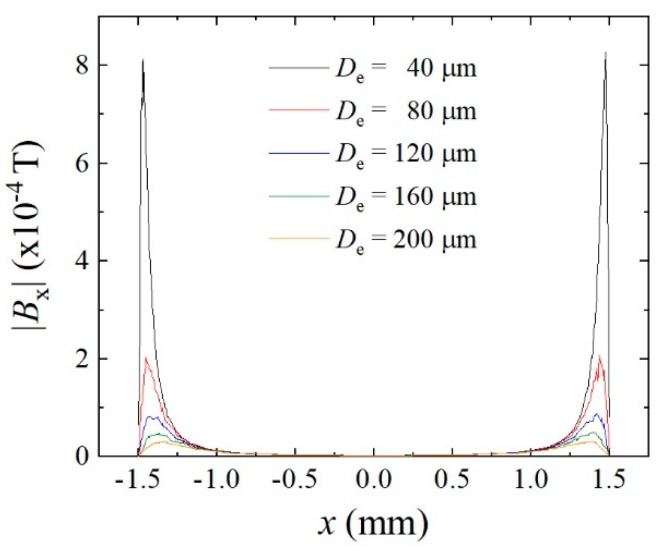
Magnetic field distributions along the axis of elements in the neighboring element position.

**Figure 19 micromachines-11-00279-f019:**
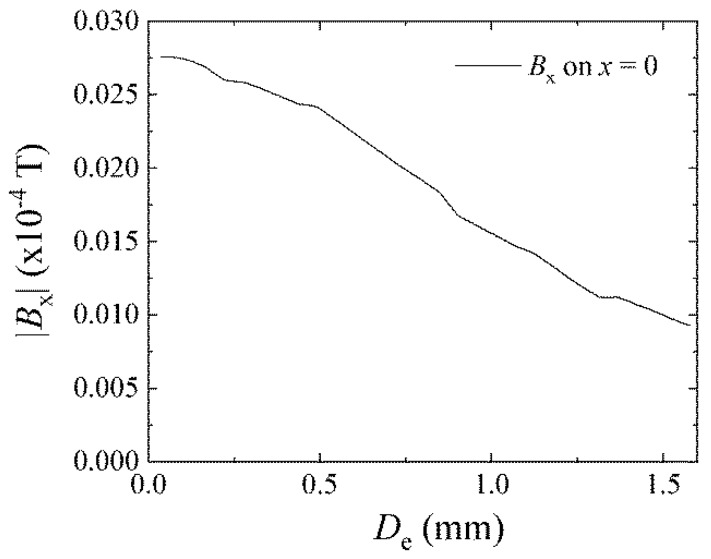
Variation of magnetic field *B_x_* as a function of the distance from the element *D_e_*.

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
