# Peer review of "Magnetic Domain Transition of Adjacent Narrow Thin Film Strips with Inclined Uniaxial Magnetic Anisotropy"

_micromachines, 2020, doi:10.3390/mi11030279_

Round 1

Reviewer 1 Report

In this work arrays of soft magnetic thin film with inclined easy axis are perpetrated and their domain structure is investigated by means of VSM and Kerr microscope. Three possible domain configurations, two single domains (up and down) and a multidomain were observed. Though the results are interesting, it is not related directly to the magnetoimpedance effect. So it should be removed from the title to not mislead the reader.
* If the individual components are bistable at high inclination angles, why there are no steps in the hysteresis loop?
* Also, the author cites only himself. There is a number of papers on GMI effect as well as on relationship of GMI effect and domain structure. For publication in WOS journal the number of self-citations must be below 30%

If the authors address aforementioned points the manuscript can be reconsidered for publication.

Reviewer 2 Report

The paper looks interesting and original. It could be published after the revision. There are some questions and doubts:

1) “The result shows that the hidden inclined Landau–Lifshitz domain appears by applying a distributed normal field as the same as an individual element. “

The main question is why “the domain transition of the individual elements which was fabricated on a same substrate occur at almost the same value of magnetic field.”

The explanation that “the gradually transition of domain is explained as a result of the magnetic mutual interaction.” – is no sufficient.

The key parameter is the distance between the elements. Authors should vary this distance and present the results for as minimum 2-3 different distances to clear the mechanism of magnetization reversal in many-body elements.

2) “A domain observation shows that the step-like change is due to a magnetization transition within three states, including the longitudinal single domain with parallel state, that with anti-parallel state and the inclined Landau–Lifshitz domain.”

It is better to say about 2 states (longitudinal and inclined) or 4 states (2 longitudinal and 2 inclined) depending on the interpretation of 2 parallel states.

3) Why “70 degree” was selected? Is it an occasional value?

4) “easy axis 71 degree”.  Are you sure that you have measured the angle so precisely?

5) Page 6, lines  53-70. Why namely these parameters of the elements distributions were selected?

6) “three different directions of easy axis were made in 61 degree, 71 degree, and 90 degree.” Why namely these angles were selected?

7) In Title “...Magneto-Impedance Element...”

There are no measurements of magneto-impedance in the paper. It is not clear why these elements are called in such a way.

8) The list of Reference looks inadequate. It should be supplemented by other authors.

Round 2

Reviewer 2 Report

The answer to my comment No1 is not satisfied.

I repeat: The distance between the elements is very important. If author can’t (or don’t want) present the results for different distance between the elements, he should, as minimum, demonstrate the understanding of the mechanism of the influence of the distance on the mutual magnetic interaction, and explain at least qualitatively, what happens with increasing or decreasing of the distance.